# On the Importance of Gradients for Detecting Distributional Shifts in the Wild

**Rui Huang**
Department of Computer Sciences
University of Wisconsin-Madison
huangrui@cs.wisc.edu

**Andrew Geng**
Department of Computer Sciences*
University of Wisconsin-Madison
ageng@wisc.edu

**Yixuan Li**
Department of Computer Sciences
University of Wisconsin-Madison
sharonli@cs.wisc.edu

## Abstract

Detecting out-of-distribution (OOD) data has become a critical component in ensuring the safe deployment of machine learning models in the real world. Existing OOD detection approaches primarily rely on the output or feature space for deriving OOD scores, while largely overlooking information from the *gradient space*. In this paper, we present `GradNorm`, a simple and effective approach for detecting OOD inputs by utilizing information extracted from the gradient space. `GradNorm` directly employs the vector norm of gradients, backpropagated from the KL divergence between the softmax output and a uniform probability distribution. Our key idea is that the magnitude of gradients is higher for in-distribution (ID) data than that for OOD data, making it informative for OOD detection. `GradNorm` demonstrates superior performance, reducing the average FPR95 by up to 16.33% compared to the previous best method. Code and data available: https://github.com/deeplearning-wisc/gradnorm_ood.

## 1 Introduction

When deploying machine learning models in the real world, there is an increasingly important question to ask: *"Is the model making a faithful prediction for something it was trained on, or is the model making an unreliable prediction for something it has not been exposed to during training?"* We desire models that are not only accurate on their familiar data distribution, but also aware of uncertainty outside the training distribution. This gives rise to the importance of out-of-distribution (OOD) detection, which determines whether an input is in-distribution (ID) or OOD. As of recently a plethora of literature has emerged to address the problem of OOD uncertainty estimation [2, 13, 14, 16, 24, 26–29, 31, 32, 41].

The main challenge in OOD uncertainty estimation stems from the fact that modern deep neural networks can easily produce overconfident predictions on OOD inputs [34]. This phenomenon makes the separation between ID and OOD data a non-trivial task. Much of the prior work focused on deriving OOD uncertainty measurements from the activation space of the neural network, e.g., using model output [13, 14, 24, 27, 29] or feature representations [26]. Yet, this leaves an alternative space—model parameter and its *gradient space*—largely unexplored. Will a model react to ID and OOD inputs differently in its gradient space, and if so, can we discover distinctive signatures to separate ID and OOD data from gradients?

---

*Work done while A.G was working as an undergraduate research assistant with Li's lab.

35th Conference on Neural Information Processing Systems (NeurIPS 2021).

In this paper, we tackle this key question by exploring and exploiting the richness of the gradient space, ultimately showing that gradients carry surprisingly useful signals for OOD detection. Formally, we present `GradNorm`, a simple and effective approach for detecting OOD inputs by utilizing gradient extracted from a pre-trained neural network. Specifically, `GradNorm` employs the vector norm of gradients directly as an OOD scoring function. Gradients are backpropagated from the Kullback-Leibler (KL) divergence [23] between the softmax output and a uniform distribution. ID data is expected to have larger KL divergence because the prediction tends to concentrate on one of the ground-truth classes and is therefore less uniformly distributed. As depicted in Figure 1, our key idea is that the gradient norm of the KL divergence is higher for ID data than that for OOD data, making it informative for OOD uncertainty estimation.

We provide both empirical and theoretical insights, demonstrating the superiority of `GradNorm` over both output-based and feature-based methods. Empirically, we establish superior performance on a large-scale ImageNet benchmark, as well as a suite of common OOD detection benchmarks. `GradNorm` outperforms the previous best method by a large margin, with up to 16.33% reduction in false-positive rate (FPR95). Theoretically, we show that `GradNorm` captures the *joint information* between the feature and the output space. The joint information results in an overall stronger separability than using either feature or output space alone.

Our **key results and contributions** are summarized as follows.

- We propose `GradNorm`, a simple and effective gradient-based OOD uncertainty estimation method, which is both label-agnostic (no label required for backpropagation) and OOD-agnostic (no outlier data required). `GradNorm` reduces the average FPR95 by 16.33% compared to the current best method.
- We perform comprehensive analyses that improve understandings of the gradient-based method under (1) different network architectures, (2) gradient norms extracted at varying depths, (3) different loss functions for backpropagation, and (4) different vector norms for aggregating gradients.
- We perform a mathematical analysis of `GradNorm` and show that it can be decomposed into two terms, jointly characterizing information from both feature and output space, which demonstrates superiority.

## 2 Preliminaries

We start by recalling the general setting of the supervised learning problem. We denote by $\mathcal{X} = \mathbb{R}^d$ the input space and $\mathcal{Y} = \{1, 2, ..., C\}$ the output space. A learner is given access to a set of training data $D = \{(\mathbf{x}_i, y_i)\}_{i=1}^N$ drawn from an unknown joint data distribution $P$ defined on $\mathcal{X} \times \mathcal{Y}$. A neural network $f(\mathbf{x}; \theta) : \mathcal{X} \rightarrow \mathbb{R}^C$ minimizes the empirical risk:

$$R_{\mathcal{L}}(f) = \mathbb{E}_D(\mathcal{L}_{\text{CE}}(f(\mathbf{x}; \theta), y)),$$

where $\theta$ is the parameters of the network, and $\mathcal{L}_{\text{CE}}$ is the commonly used cross-entropy loss:

$$\mathcal{L}_{\text{CE}}(f(\mathbf{x}), y) = -\log \frac{e^{f_y(\mathbf{x})/T}}{\sum_{c=1}^C e^{f_c(\mathbf{x})/T}}. \tag{1}$$

Specifically, $f_y(\mathbf{x})$ denotes the $y$-th element of $f(\mathbf{x})$ corresponding to the ground-truth label $y$, and $T$ is the temperature.

**Problem statement** Out-of-distribution (OOD) detection can be formulated as a binary classification problem. In practice, OOD is often defined by a distribution that simulates unknowns encountered during deployment time, such as samples from an irrelevant distribution whose label set has no intersection with $\mathcal{Y}$ and *therefore should not be predicted by the model*. Given a classifier $f$ learned on training samples from in-distribution $P$, the goal is to design a binary function estimator,

$$g(\mathbf{x}) = \begin{cases} \text{in}, & \text{if } S(\mathbf{x}) \geq \gamma \\ \text{out}, & \text{if } S(\mathbf{x}) < \gamma, \end{cases}$$

that classifies whether a sample $\mathbf{x} \in \mathcal{X}$ is from $P$ or not. $\gamma$ is commonly chosen so that a high fraction (e.g., 95%) of ID data is correctly classified. The key challenge is to derive a scoring function $S(\mathbf{x})$ that captures OOD uncertainty. Previous approaches have primarily relied on the model's output or features for OOD uncertainty estimation. Instead, our approach seeks to compute $S(\mathbf{x})$ based on the information extracted from the *gradient space*, which we describe in the next section.

# 3 Gradient-based OOD Detection

In this section, we describe our method `GradNorm`. We start by introducing the loss function for backpropagation and then describe how to leverage the gradient norm for OOD uncertainty estimation.

We calculate gradients *w.r.t.* each parameter by backpropagating the Kullback-Leibler (KL) divergence [23] between the softmax output and a uniform distribution. Formally, KL divergence quantifies how close a model-predicted distribution $q = \{q_i\}$ is to a reference probability distribution $p = \{p_i\}$,

$$D_{\mathrm{KL}}(p \,\|\, q) = \sum_i p_i \log \frac{p_i}{q_i} = -\sum_i p_i \log q_i + \sum_i p_i \log p_i = H(p, q) - H(p). \tag{2}$$

In particular, we set the reference distribution to be uniform $\mathbf{u} = [1/C, 1/C, ..., 1/C] \in \mathbb{R}^C$. The predictive probability distribution is the softmax output. Our KL divergence for backpropagation can be written as:

$$D_{\mathrm{KL}}(\mathbf{u} \,\|\, \mathrm{softmax}(f(\mathbf{x}))) = -\frac{1}{C} \sum_{c=1}^{C} \log \frac{e^{f_c(\mathbf{x})/T}}{\sum_{j=1}^{C} e^{f_j(\mathbf{x})/T}} - H(\mathbf{u}), \tag{3}$$

where the first term is the cross-entropy loss between the softmax output and a uniform vector $\mathbf{u}$, and the second term $H(\mathbf{u})$ is a constant. The KL divergence measures how much the predictive distribution is away from the uniform distribution. Intuitively, ID data is expected to have larger KL divergence because the prediction tends to concentrate on the ground-truth class and is thus distributed less uniformly.

**GradNorm as OOD score** For a given parameter $w$, the gradient of the above KL divergence is:

$$\frac{\partial D_{\mathrm{KL}}(\mathbf{u} \,\|\, \mathrm{softmax}(f(\mathbf{x})))}{\partial w} = \frac{1}{C} \sum_{i=1}^{C} \frac{\partial \mathcal{L}_{\mathrm{CE}}(f(\mathbf{x}), i)}{\partial w}, \tag{4}$$

where $w$ is a component of network parameter $\theta$. Notice that the gradient of the entropy term is 0, *i.e.* $\partial H(\mathbf{u})/\partial w = 0$. In other words, the gradient of KL divergence is equivalent to averaging the derivative of the categorical cross-entropy loss for *all* labels.

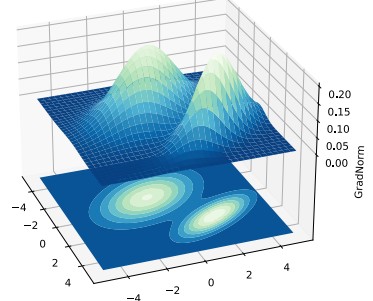

Figure 1: An example of two-dimensional input space. Input data is depicted in the $xy$-plane, while gradient norm for each input is depicted in the $z$-dimension. The magnitude of gradients is higher for ID data (light green) than that for OOD data (deep blue).

We now define the OOD score via a vector norm of gradients of the selected parameters:

$$S(\mathbf{x}) = \|\frac{\partial D_{\mathrm{KL}}(\mathbf{u} \,\|\, \mathrm{softmax}(f(\mathbf{x}))}{\partial \mathbf{w}}\|_p, \tag{5}$$

where $\|\cdot\|_p$ denotes $L_p$-norm and $\mathbf{w}$ is the set of parameters in vector form[2]. We term our method `GradNorm`, short for gradient norm. In practice, `GradNorm` can be conveniently implemented by calculating the cross-entropy loss between the predicted softmax probability and a uniform vector as the target. We will discuss the choices and impacts of the selected parameter set $\mathbf{w}$ in Section 4.2.

**Rationale of GradNorm** Our operating hypothesis is that using the KL divergence for backpropagation, the gradient norm is higher for ID data than that for OOD data. As we show in Section 4.2, using the gradient norm of the KL divergence is more effective than using the KL divergence directly. Moreover, `GradNorm` derived from the KL divergence with a uniform target offers two advantages over gradient norms derived from the standard cross-entropy loss.

- First, our method is *label-agnostic* and does not require any ground-truth label. It can be flexibly used during inference time when the label is unavailable for either ID or OOD data.
- Second, it captures the *uncertainty across all categories*, providing more information for OOD detection. We will provide empirical evidence to show the importance of utilizing all labels in Section 4.2.

---

[2]We concatenate all selected parameters into a single vector regardless of the original shapes of the parameters.

| Method Space | Method | iNaturalist | | SUN | | Places | | Textures | | Average | |
|---|---|---|---|---|---|---|---|---|---|---|---|
| | | FPR95 ↓ | AUROC ↑ | FPR95 ↓ | AUROC ↑ | FPR95 ↓ | AUROC ↑ | FPR95 ↓ | AUROC ↑ | FPR95 ↓ | AUROC ↑ |
| Output | MSP [13] | 63.69 | 87.59 | 79.98 | 78.34 | 81.44 | 76.76 | 82.73 | 74.45 | 76.96 | 79.29 |
| | ODIN [27] | 62.69 | 89.36 | 71.67 | 83.92 | 76.27 | 80.67 | 81.31 | 76.30 | 72.99 | 82.56 |
| | Energy [29] | 64.91 | 88.48 | 65.33 | 85.32 | 73.02 | 81.37 | 80.87 | 75.79 | 71.03 | 82.74 |
| Feature | Mahalanobis [26] | 96.34 | 46.33 | 88.43 | 65.20 | 89.75 | 64.46 | **52.23** | 72.10 | 81.69 | 62.02 |
| Gradient | **GradNorm (ours)** | **50.03** | **90.33** | **46.48** | **89.03** | **60.86** | **84.82** | 61.42 | **81.07** | **54.70** | **86.31** |

Table 1: **Main Results.** OOD detection performance comparison between `GradNorm` and baselines. All methods utilize the standard ResNetv2-101 model trained on ImageNet [7]. The classification model is trained on ID data only. ↑ indicates larger values are better, while ↓ indicates smaller values are better. All values are percentages. All methods are post hoc and can be directly used for pre-trained models.

## 4 Experiments

In this section, we evaluate `GradNorm` on a large-scale OOD detection benchmark with ImageNet-1k as in-distribution dataset [16]. We describe experimental setup in Section 4.1 and demonstrate the superior performance of `GradNorm` over existing approaches in Section 4.2, followed by extensive ablations and analyses that improve the understandings of our approach.

### 4.1 Experimental Setup

**Dataset** We use the large-scale ImageNet OOD detection benchmark proposed by Huang and Li [16]. ImageNet benchmark is not only more realistic (with higher resolution images) but also more challenging (with a larger label space of 1,000 categories). We evaluate on four OOD test datasets, which are from subsets of `iNaturalist` [43], `SUN` [47], `Places` [51], and `Textures` [5], with non-overlapping categories *w.r.t.* ImageNet-1k (see Appendix B.1 for detail). The evaluations span a diverse range of domains including fine-grained images, scene images, and textural images. We further evaluate on CIFAR benchmarks that are routinely used in literature (see Appendix A).

**Model and hyperparameters** We use Google BiT-S models[3] [21] pre-trained on ImageNet-1k with a ResNetv2-101 architecture [11]. We report performance on an alternative architecture, DenseNet-121 [15], in Section 4.2. Additionally, we use $L_1$-norm-based OOD scores as the default and explore the effect of other $L_p$-norms in Section 4.2. The temperature parameter $T$ is set to be 1 unless specified otherwise, and we explore the effect of different temperatures in Section 4.2. At test time, all images are resized to $480 \times 480$.

### 4.2 Results and Ablation Studies

**Comparison with output- and feature-based methods** The results for ImageNet evaluations are shown in Table 1, where `GradNorm` demonstrates superior performance. We report OOD detection performance for each OOD test dataset, as well as the average over the four datasets. For a fair comparison, all the methods use the same pre-trained backbone, without regularizing with auxiliary outlier data. In particular, we compare with MSP [13], ODIN [27], Mahalanobis [26], as well as Energy [29]. *Details and hyperparameters of baseline methods can be found in Appendix B.2.*

`GradNorm` outperforms the best output-based baseline, Energy score [29], by **16.33**% in FPR95. `GradNorm` also outperforms a competitive feature-based method, Mahalanobis [26], by **26.99**% in FPR95. We hypothesize that the increased size of label space makes the class-conditional Gaussian density estimation less viable. It is also worth noting that significant overheads can be introduced by some methods. For instance, Mahalanobis [26] requires collecting feature representations from intermediate layers over the entire training set, which is expensive for large-scale datasets such as ImageNet. In contrast, `GradNorm` can be conveniently used through a simple gradient calculation without hyper-parameter tuning or additional training.

**Gradients from the last layer is sufficiently informative** In this ablation, we investigate several variants of `GradNorm` where the gradients are extracted from different network depths. Specifically, we consider gradients of (1) **block n**: all trainable parameters in the $n$-th block, (2) **all parameters**: all trainable parameters from all layers of the network, and (3) **last layer parameters**: weight parameters from the last fully connected (FC) layer.

---

[3] https://github.com/google-research/big_transfer

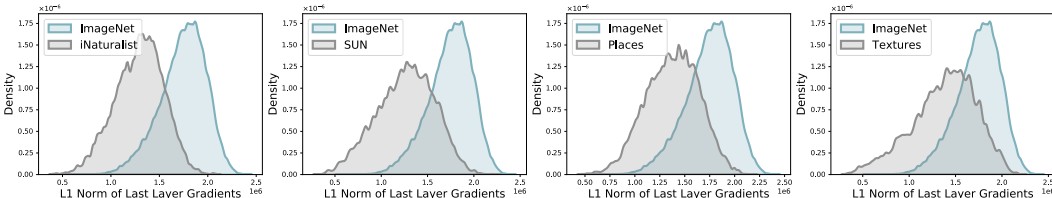

(a) Gradient norms using KL divergence between the softmax prediction and the **uniform** target.

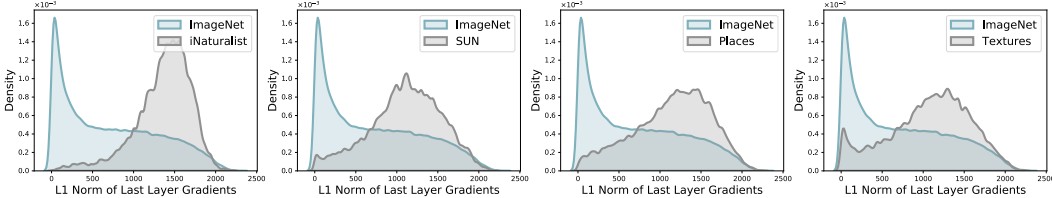

(b) Gradient norms using KL divergence between the softmax prediction and the **one-hot** target.

Figure 2: Comparison of $L_1$-norm distributions of last layer gradients between KL divergence with *uniform* target and KL divergence with *one-hot* target. We show in-distribution data in green and OOD data in gray.

Table 2 contrasts the OOD detection performance using different *gradient space*. For each setting, we report the FPR95 and AUROC averaged across four OOD datasets. We observe that gradients from deeper layers tend to yield significantly better performance than shallower layers. This is desirable since gradients *w.r.t.* deeper layers are computationally more efficient than shallower layers. Interestingly, GradNorm obtained from the last linear layer yield the best results among all variants. Practically, one only needs to perform backpropagation *w.r.t.* the last linear layer, which incurs negligible computations. Therefore, our main results are based on the norm of gradients extracted from weight parameters in the last FC layer of the neural network.

| Gradient Space | FPR95 ↓ | AUROC ↑ |
|---|---|---|
| Block 1 | 73.52 | 76.41 |
| Block 2 | 74.34 | 76.63 |
| Block 3 | 71.73 | 78.11 |
| Block 4 | 65.07 | 85.11 |
| All params | 69.35 | 81.14 |
| Last layer params | **54.70** | **86.31** |

Table 2: Effect of GradNorm using different subset of gradients. Gradient norm derived from deeper layers yield better OOD detection performance.

**GradNorm with one-hot v.s. uniform targets**   In this ablation, we contrast GradNorm derived using uniform targets (ours) v.s. one-hot targets. Specifically, our scoring function Equation 5 is equivalent to

$$S(\mathbf{x}) = \|\frac{1}{C}\sum_{i=1}^{C}\frac{\partial \mathcal{L}_{\text{CE}}(f(\mathbf{x}), i)}{\partial \mathbf{w}}\|, \tag{6}$$

which captures the gradient of cross-entropy loss across *all* labels. In contrast, we compare against an alternative scoring function that utilizes *only one* dominant class label:

$$S_{\text{one-hot}}(\mathbf{x}) = \|\frac{\partial \mathcal{L}_{\text{CE}}(f(\mathbf{x}), \hat{y})}{\partial \mathbf{w}}\|, \tag{7}$$

where $\hat{y}$ is the predicted class with the largest output.

We first analyze the score distributions using uniform targets (**top**) and one-hot targets (**bottom**) for ID and OOD data in Figure 2. There are two salient observations we can draw: (1) using uniform target (ours), gradients of ID data indeed have larger magnitudes than those of OOD data, as the softmax prediction tends to be less uniformly distributed (and therefore results in a larger KL divergence). In contrast, the gradient norm using one-hot targets shows the opposite trend, with ID data having lower magnitudes. This is also expected since the training objective explicitly minimizes the cross-entropy loss, which results in smaller gradients for the majority of ID data. (2) The score distribution using one-hot targets displays a strong overlapping between ID (green) and OOD (gray) data, with large variances. In contrast, our method GradNorm can significantly improve the separability between ID and OOD data, resulting in better OOD detection performance.

Figure 3 reports the OOD detection performance using uniform targets (ours) v.s. one-hot targets. We use $L_1$-norm in both cases. For one-hot targets, we use the negative norm, *i.e.* $-S_{\text{one-hot}}(\mathbf{x})$, to align with the convention that ID data has higher scores. GradNorm with uniform targets outperforms its counterpart with one-hot targets by a large margin. For instance, GradNorm reduces FPR95 by **48.35**% when evaluated on the SUN dataset. Our analysis signifies the importance of measuring OOD uncertainty using all label information.

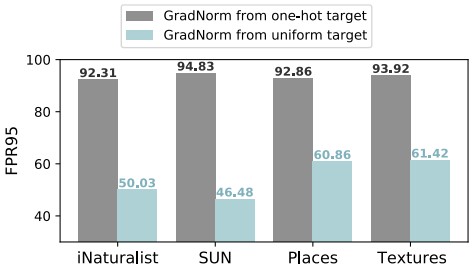

Figure 3: OOD detection performance (FPR95) comparison between uniform (ours) v.s. one-hot target.

**GradNorm is effective on alternative neural network architecture**  We evaluate GradNorm on a different architecture DenseNet-121 [15], and report performance in Table 3. GradNorm is consistently effective, outperforming the best baseline, Energy [29], by **10.29**% in FPR95.

| Method Space | Method | iNaturalist | | SUN | | Places | | Textures | | Average | |
|---|---|---|---|---|---|---|---|---|---|---|---|
| | | FPR95 ↓ | AUROC ↑ | FPR95 ↓ | AUROC ↑ | FPR95 ↓ | AUROC ↑ | FPR95 ↓ | AUROC ↑ | FPR95 ↓ | AUROC ↑ |
| Output | MSP [13] | 48.55 | 89.16 | 69.39 | 80.46 | 71.42 | 80.11 | 68.51 | 78.69 | 64.47 | 82.11 |
| | ODIN [27] | 37.00 | 93.29 | 57.30 | 86.12 | 61.91 | 84.14 | 56.49 | 84.62 | 53.18 | 87.04 |
| | Energy [29] | 36.39 | 93.29 | 54.91 | 86.53 | 59.98 | **84.29** | 53.87 | 85.07 | 51.29 | 87.30 |
| Feature | Mahalanobis [26] | 97.36 | 42.24 | 98.24 | 41.17 | 97.32 | 47.27 | 62.78 | 56.53 | 88.93 | 46.80 |
| Gradient | **GradNorm (ours)** | **23.87** | **93.97** | **43.04** | **87.79** | **53.92** | 83.04 | **43.16** | **87.48** | **41.00** | **88.07** |

Table 3: OOD detection performance comparison on a different architecture, **DenseNet-121** [15]. Model is trained on ImageNet-1k [7] as the ID dataset. All methods are post hoc and can be directly used for pre-trained models.

$L_1$**-norm is the most effective**  How does the choice of $L_p$-norm in Equation 5 affect the OOD detection performance? To understand this, we show in Figure 4 the comparison using $L_{1\sim4}$-norm, $L_\infty$-norm, as well as the fraction norm (with $p = 0.3$). Compared with higher-order norms, $L_1$-norm achieves the best OOD detection performance on all four datasets. We hypothesize that $L_1$-norm is better suited since it captures information equally from all dimensions in the gradient space, whereas higher-order norms will unfairly highlight larger elements rather than smaller elements (due to the effect of the exponent $p$). In the extreme case, $L_\infty$-norm only considers the largest element (in absolute value) and results in the worst OOD detection performance among all norms. On the other hand, the fraction norm overall does not outperform $L_1$-norm. *We additionally provide results for more $L_p$-norms in Appendix C with $p = \{0.3, 0.5, 0.8, 1, 2, 3, 4, 5, 6, \infty\}$.*

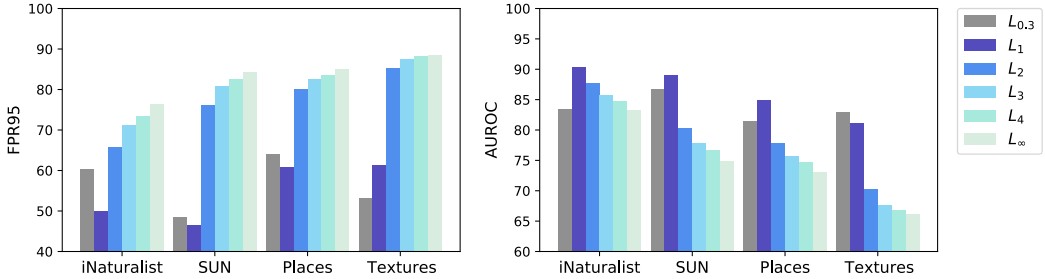

Figure 4: OOD detection performance comparison under different $L_p$-norms. We show FPR95 (*left*) and AUROC (*right*).

**Effect of temperature scaling** We evaluate our method GradNorm with different temperatures $T$ from $T = 0.5$ to $T = 1024$. As shown in Figure 5, $T = 1$ is optimal, while either increasing or decreasing the temperature will degrade the performance. This can be explained mathematically via the $V$ term in Equation 9. Specifically, using a large temperature will result in a smoother softmax distribution, with $C \cdot \frac{e^{f_j/T}}{\sum_{j=1}^{C} e^{f_j/T}}$ closer to 1 (and $V \rightarrow 0$). This leads to a less distinguishable

distribution between ID and OOD. Our method can be hyperparameter-free by setting $T = 1$. For completeness, we have included numerical results under a wider range of $T$ in Appendix D.

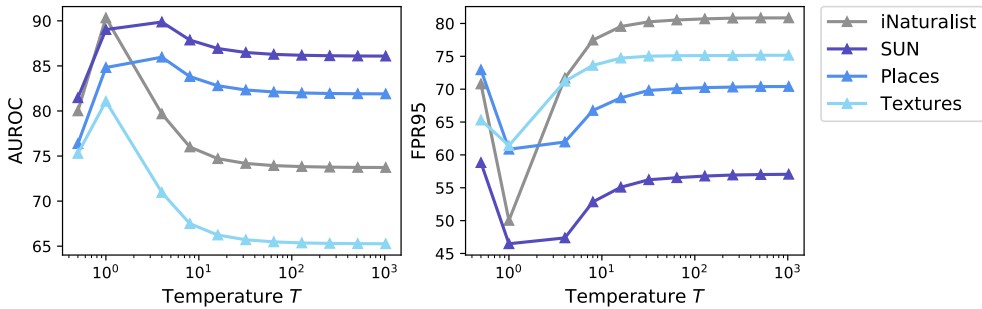

Figure 5: OOD detection performance of `GradNorm` with varying temperature parameter $T$. We show AUROC (*left*) and FPR95 (*right*).

**GradNorm is more effective than directly using KL divergence** We provide an ablation, contrasting the performance of using `GradNorm` v.s. using the KL divergence derived from Equation 3 as OOD scoring function. The results are show in Figure 6, where `GradNorm` yields significantly better performance than the KL divergence directly extracted from the *output space*, demonstrating the superiority of *gradient space* for OOD detection.

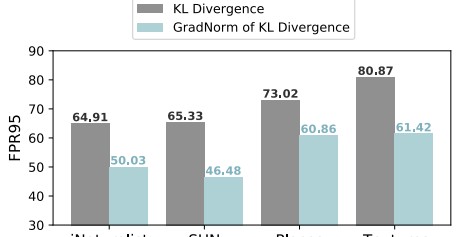

Figure 6: Comparison between `GradNorm` v.s. directly using the KL divergence as scoring function.

**Effect of model capacity** In this ablation, we explore the OOD detection performance of `GradNorm` with varying model capacities. For the ease of experiments, we directly use Google BiT-S models pre-trained on ImageNet-1k [7]. We compare the performance of the following model family (in increasing size): BiT-S-R50x1, BiT-S-R101x1, BiT-S-R50x3, BiT-S-R152x2, BiT-S-R101x3. All models are ResNetv2 architectures with varying depths and width factors. The average performance on 4 OOD datasets is reported in Table 4. OOD detection performance is optimal when the model size is relatively small

| Model Size (depth x width) | FPR95 ↓ | AUROC ↑ |
|---|---|---|
| 50x1 | 56.91 | 84.17 |
| 101x1 | **55.84** | **84.63** |
| 50x3 | 61.74 | 81.89 |
| 152x2 | 61.76 | 81.33 |
| 101x3 | 66.20 | 78.89 |

Table 4: OOD detection performance as the model capacity increases.

(ResNetv2-101x1), while further increasing model capacity will degrade the performance. Our experiments suggest that overparameterization can make gradients less distinguishable between ID and OOD data and that `GradNorm` is more suitable under a mild model capacity.

## 5 Analysis of Gradient-based Method

In this section, we analyze the best variant of `GradNorm`, $L_1$-norm of the last layer gradients (see Section 4.2), and provide insights on the mathematical interpretations. Specifically, we denote the last FC layer in a neural network by:

$$f(\mathbf{x}) = \mathbf{W}^\top \mathbf{x} + \mathbf{b}, \tag{8}$$

where $f = [f_1, f_2, \ldots, f_C]^\top \in \mathbb{R}^C$ is the logit output, $\mathbf{x} = [x_1, x_2, \ldots, x_m]^\top \in \mathbb{R}^m$ is the input feature vector, $\mathbf{W} \in \mathbb{R}^{m \times C}$ is the weight matrix, and $\mathbf{b} \in \mathbb{R}^C$ is the bias vector.

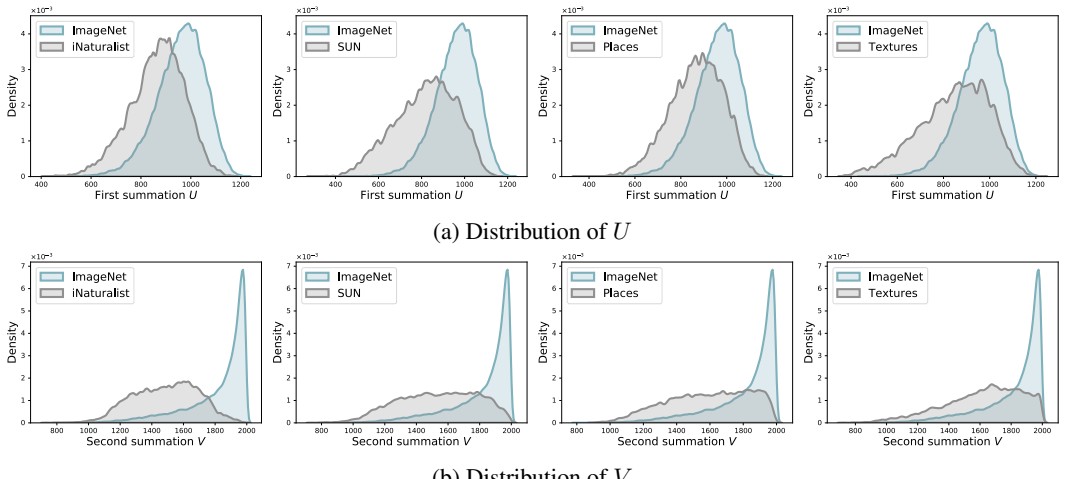

(a) Distribution of $U$

(b) Distribution of $V$

Figure 7: We show the distributions of the two summations decomposed from the $L_1$-norm of the last layer gradient, for both in-distribution data (blue) and out-of-distribution data (gray).

**GradNorm captures joint information between feature and output** First we can rewrite the KL divergence between the softmax prediction and the uniform target as:

$$D_{\text{KL}}(\mathbf{u}\|\text{softmax}(f(\mathbf{x}))) = -\frac{1}{C}\sum_{c=1}^{C}\log\frac{e^{f_c/T}}{\sum_{j=1}^{C}e^{f_j/T}} - H(\mathbf{u})$$

$$= -\frac{1}{C}\left(\frac{1}{T}\sum_{c=1}^{C}f_c - C\cdot\log\sum_{j=1}^{C}e^{f_j/T}\right) - H(\mathbf{u}).$$

Then we consider the derivative of $D_{\text{KL}}$ *w.r.t.* each output logit $f_c$:

$$\frac{\partial D_{\text{KL}}}{\partial f_c} = -\frac{1}{CT}\left(1 - CT\cdot\frac{\partial\left(\log\sum_{j=1}^{C}e^{f_j/T}\right)}{\partial f_c}\right)$$

$$= -\frac{1}{CT}\left(1 - C\cdot\frac{e^{f_c/T}}{\sum_{j=1}^{C}e^{f_j/T}}\right).$$

Next the derivative of $D_{\text{KL}}$ *w.r.t.* the weight matrix can be written as:

$$\frac{\partial D_{\text{KL}}}{\partial\mathbf{W}} = \mathbf{x}\frac{\partial D_{\text{KL}}}{\partial f} = -\frac{1}{CT}\cdot[x_1, x_2, \ldots, x_m]^{\top}[1 - C\cdot\frac{e^{f_1/T}}{\sum_{j=1}^{C}e^{f_j/T}}, \ldots, 1 - C\cdot\frac{e^{f_C/T}}{\sum_{j=1}^{C}e^{f_j/T}}].$$

Finally, the $L_1$-norm of gradients of the weight matrix is simply the sum of absolute values of all elements in the gradient matrix:

$$S(\mathbf{x}) = \sum_{i=1}^{m}\sum_{j=1}^{C}\left|\left(\frac{\partial D_{\text{KL}}}{\partial\mathbf{W}}\right)_{ij}\right| = \frac{1}{CT}\sum_{i=1}^{m}\left(|x_i|\left(\sum_{j=1}^{C}\left|1 - C\cdot\frac{e^{f_j/T}}{\sum_{j=1}^{C}e^{f_j/T}}\right|\right)\right)$$

$$= \frac{1}{CT}\left(\sum_{i=1}^{m}|x_i|\right)\left(\sum_{j=1}^{C}\left|1 - C\cdot\frac{e^{f_j/T}}{\sum_{j=1}^{C}e^{f_j/T}}\right|\right) \qquad (9)$$

$$\triangleq \frac{1}{CT}U\cdot V,$$

where the first multiplicative term $U = \sum_{i=1}^{m}|x_i|$ is the $L_1$-norm of the feature vector $\mathbf{x}$, and the second term $V$ characterizes information in the output space.

**Ablation on $U$ and $V$** In Figure 7 we plot distribution densities of $U$ and $V$, for both ID and OOD data. It is important to note that $U$ and $V$ measure statistical distributions in the feature space and the output space, respectively. Therefore, GradNorm captures the joint information between the feature and the output space. The multiplication of both $U$ and $V$ results in an overall stronger separability between ID and OOD, as seen in Figure 2a. We report the OOD detection performance using $U$ and $V$ individually as scoring functions in Table 5, both of which are less competitive than GradNorm.

| Method | iNaturalist | | SUN | | Places | | Textures | | Average | |
|---|---|---|---|---|---|---|---|---|---|---|
| | FPR95 ↓ | AUROC ↑ | FPR95 ↓ | AUROC ↑ | FPR95 ↓ | AUROC ↑ | FPR95 ↓ | AUROC ↑ | FPR95 ↓ | AUROC ↑ |
| U (feature space) | 77.84 | 74.33 | 61.90 | 78.74 | 76.42 | 72.75 | 67.84 | 72.77 | 71.00 | 74.65 |
| V (output space) | 66.14 | 88.45 | 69.49 | 83.13 | 75.95 | 78.98 | 81.13 | 76.06 | 73.18 | 81.66 |
| **U · V (joint space)** | **50.05** | **90.33** | **46.48** | **89.03** | **60.86** | **84.82** | **61.42** | **81.07** | **54.70** | **86.31** |

Table 5: OOD detection performance using the decomposed $U$ (feature space) and $V$ (output space) as scoring functions. Model is ResNetv2-101 trained on ImageNet-1k [7].

# 6 Discussion

To the best of our knowledge, there is very limited prior work studying how to use gradients for OOD detection. In this section, we discuss connections and differences between GradNorm and previous OOD detection approaches that utilize gradient information, in particular ODIN (Section 6.1) and Lee and AlRegib's approach (Section 6.2).

## 6.1 Comparison with ODIN

Our work is inspired by ODIN [27], which first explored using gradient information for OOD detection. In particular, ODIN proposed using input pre-processing by adding small perturbations obtained from the input gradients. The goal of ODIN perturbations is to increase the softmax score of any given input by reinforcing the model's belief in the predicted label. Ultimately the perturbations have been found to create a greater gap between the softmax scores of ID and OOD inputs, thus making them more separable and improving the performance of OOD detection.

It is important to note that ODIN only uses gradients *implicitly* through input perturbation, and OOD scores are still derived from the output space of the perturbed inputs. Different from ODIN, GradNorm utilizes information solely obtained from the gradient space. The effectiveness of GradNorm beckons a revisiting of combining information obtainable from the gradient space and the output space, which could provide a stronger method. We leave this question for future exploration.

## 6.2 Comparison with Lee and AlRegib

Lee and AlRegib [25] proposed to train an auxiliary binary classifier using gradient information from ID and OOD data. Importantly, they do not directly use gradient norms for OOD detection, but instead, use them as the input for training a separate binary classifier. Furthermore, the binary classifier is trained on the OOD datasets, which can unfairly overfit the test data and does not suit OOD-agnostic settings in the real world. In contrast, our methodology mitigates the shortcomings in that GradNorm (1) does not require any new model training, (2) is hyperparameter-free, and (3) is suitable for OOD-agnostic settings. For these reasons, these two methods are not directly comparable. However, for completeness, we also reproduce Lee and AlRegib's method using random noise as a surrogate of OOD data and compare it with GradNorm in Appendix E. GradNorm outperforms their approach by 15.45% in FPR95 in this fair comparison.

Moreover, we provide comprehensive ablation studies and analyses on different design choices in using gradient-based methods for OOD detection (network architectures, gradients at different layers, loss functions for backpropagation, different $L_p$-norms, diverse evaluation datasets, and different temperatures, etc.), which were previously not studied in [25]. In particular, Lee and AlRegib utilize $L_2$-norm gradients without comparing them with other norms. Our ablation study leads to the new finding that $L_1$-norm works best among all variants with GradNorm, and outperforms $L_2$-norm by up to 22.31% in FPR95. Furthermore, Lee and AlRegib utilize gradients from all layers to train a separate binary classifier, which can cause the computational cost to become intractable for deeper and larger models. In contrast, with GradNorm we show that the last layer gradient will always yield the best performance among all gradient set selections. Consequently, GradNorm incurs negligible computational cost. We believe such thorough understandings will be valuable for the field.

## 7 Related Work

**OOD uncertainty estimation with discriminative models** The problem of classification with rejection can date back to early works on abstention [3, 9], which considered simple model families such as SVMs [6]. A comprehensive survey on OOD detection can be found in [49]. We highlight representative works for post hoc detection. The phenomenon of neural networks' overconfidence to OOD data is revealed by Nguyen et al. [34]. Early works attempted to improve the OOD uncertainty estimation by proposing the ODIN score [27], OpenMax [1], and Mahalanobis distance [26]. Recent work by Liu et al. [29] proposed using an energy score for OOD uncertainty estimation, which can be easily derived from a discriminative classifier and demonstrated advantages over the softmax confidence score both empirically and theoretically. Wang et al. [44] further showed an energy-based approach can improve OOD uncertainty estimation for multi-label classification networks. Huang and Li [16] revealed that approaches developed for common CIFAR benchmarks might not translate effectively into a large-scale ImageNet benchmark, highlighting the need to evaluate OOD uncertainty estimation in a large-scale real-world setting. Existing approaches derive OOD scores from either output or feature space. In contrast, we show that *gradient space* carries surprisingly useful information for OOD uncertainty estimation, which was underexplored in the literature.

**OOD uncertainty estimation with generative models** Alternative approaches for detecting OOD inputs resort to generative models that directly estimate density [8, 17, 18, 35, 38, 42]. An input is deemed as OOD if it lies in the low-likelihood regions. A plethora of literature has emerged to utilize generative models for OOD detection [19, 37, 39, 40, 45, 46, 48]. Interestingly, Nalisnick et al. [32] showed that deep generative models can assign a high likelihood to OOD data. Moreover, generative models can be prohibitively challenging to train and optimize, and the performance can often lag behind the discriminative counterpart. In contrast, our method relies on a discriminative classifier, which is easier to optimize and achieves stronger performance.

**Distributional shifts** Distributional shifts have attracted increasing research interests [20]. It is important to recognize and differentiate various types of distributional shift problems. Literature in OOD detection is commonly concerned about model reliability and detection of label-space shifts [13, 27, 29], where the OOD inputs have disjoint labels *w.r.t.* ID data and therefore *should not be predicted by the model*. Meanwhile, some works considered covariate shifts in the input space [30, 36], where inputs can be corruption-shifted or domain-shifted [14]. However, covariate shifts are commonly used to evaluate model robustness [12] and domain generalization performance [52], where the label space $\mathcal{Y}$ remains the same during test time. It is important to note that our work focuses on the detection of shifts where the model should not make any prediction, instead of covariate shifts where the model is expected to *generalize*.

## 8 Conclusion

In this paper, we propose `GradNorm`, a novel OOD uncertainty estimation approach utilizing information extracted from the *gradient space*. Experimental results show that our gradient-based method can improve the performance of OOD detection by up to 16.33% in FPR95, establishing superior performance. Extensive ablations provide further understandings of our approach. We hope that our research brings to light the informativeness of gradient space, and inspires future work to utilize gradient space for OOD uncertainty estimation.

## 9 Societal Impact

Our project aims to improve the reliability and safety of modern machine learning models. This stands to benefit a wide range of fields and societal activities. We believe out-of-distribution uncertainty estimation is an increasingly critical component of systems that range from consumer and business applications (e.g., digital content understanding) to transportation (e.g., driver assistance systems and autonomous vehicles), and to health care (e.g., unseen disease identification). Through this work and by releasing our code, we hope to provide machine learning researchers with a new methodological perspective and offer machine learning practitioners an easy-to-use tool that renders safety against OOD data in the real world. While we do not anticipate any negative consequences to our work, we hope to continue to build on our framework in future work.

### Acknowledgement

Research is supported by the Office of the Vice Chancellor for Research and Graduate Education (OVCRGE) with funding from the Wisconsin Alumni Research Foundation (WARF).

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
