# Supplementary Material

## A    Evaluation on CIFAR Benchmarks

**Setup** We additionally evaluate `GradNorm` on a common benchmark with CIFAR-10 and CIFAR-100 [22] as ID datasets, which is routinely used in literature [13, 27, 14, 29, 26]. We use the standard split with 50,000 training images and 10,000 test images. We pre-train a ResNet-20 network for 100 epochs. The learning rate is initially 0.1, and decays by a factor of 10 at epochs 50, 75 and 90 respectively. We evaluate on four common OOD benchmark datasets: `SVHN` [33], `LSUN` (crop) [50], `Places365` [51], and `Textures` [4].

**Results** We summarize the results in Table 6, where `GradNorm` remains competitive. In particular, `GradNorm` reduces the average FPR95 by **8.77%** on CIFAR-10 compared to the best baseline. On CIFAR-100, `GradNorm` outperforms the best baseline energy score [29] by **14.47%** in FPR95. Together with our large-scale evaluation in Section 4.2, `GradNorm` overall demonstrates superior performance compared to competitive methods in literature. While some baselines may require validation datasets, `GradNorm` is hyperparameter-free and can be used in OOD-agnostic setting. Experimental details and hyperparameters of baseline methods can be found in Appendix B.2.

| ID Data | Method | SVHN | | LSUN (crop) | | Places365 | | Textures | | Average | |
|---|---|---|---|---|---|---|---|---|---|---|---|
| | | FPR95 ↓ | AUROC ↑ | FPR95 ↓ | AUROC ↑ | FPR95 ↓ | AUROC ↑ | FPR95 ↓ | AUROC ↑ | FPR95 ↓ | AUROC ↑ |
| | MSP [13] | 66.09 | 89.86 | 37.73 | 94.87 | 70.05 | 85.99 | 68.23 | 87.62 | 60.53 | 89.59 |
| | ODIN [27] | 55.52 | 89.63 | 2.32 | 99.39 | 45.86 | 90.81 | 52.78 | 89.99 | 39.12 | 92.46 |
| CIFAR-10 | Energy [29] | 49.80 | 91.97 | 3.86 | 99.03 | 46.48 | 90.55 | 58.67 | 88.79 | 39.70 | 92.59 |
| | Mahalanobis [26] | 20.91 | 95.99 | 9.66 | 97.90 | 89.24 | 61.15 | 28.83 | 92.31 | 37.16 | 86.84 |
| | **GradNorm (ours)** | 17.76 | 96.66 | 0.23 | 99.87 | 57.85 | 85.20 | 37.71 | 90.76 | **28.39** | **93.12** |
| | MSP [13] | 86.33 | 72.56 | 66.33 | 82.06 | 87.57 | 69.05 | 90.64 | 64.02 | 82.72 | 71.92 |
| | ODIN [27] | 94.80 | 66.85 | 26.14 | 95.09 | 82.57 | 72.90 | 89.91 | 66.35 | 73.36 | 75.30 |
| CIFAR-100 | Energy [29] | 89.03 | 76.42 | 21.90 | 95.90 | 82.55 | 72.98 | 88.81 | 66.74 | 70.57 | 78.01 |
| | Mahalanobis [26] | 81.46 | 81.71 | 68.97 | 90.74 | 96.50 | 50.35 | 42.71 | 87.48 | 72.41 | 77.57 |
| | **GradNorm (ours)** | 76.77 | 79.35 | 1.12 | 99.69 | 88.74 | 65.99 | 57.75 | 81.83 | **56.10** | **81.72** |

Table 6: OOD detection performance on CIFAR-10 and CIFAR-100 benchmark. All methods utilize the standard ResNet-20 architecture.

## B    Details of Experiments

### B.1    Datasets

**Large-scale evaluation**    We use ImageNet-1k [7] as the in-distribution dataset, and evaluate on four OOD test datasets following the setup in [16]:

- **iNaturalist** [43] contains 859,000 plant and animal images across over 5,000 different species. Each image is resized to have a max dimension of 800 pixels. We evaluate on 10,000 images randomly sampled from 110 classes that are disjoint from ImageNet-1k.
- **SUN** [47] contains over 130,000 images of scenes spanning 397 categories. SUN and ImageNet-1k have overlapping categories. We evaluate on 10,000 images randomly sampled from 50 classes that are disjoint from ImageNet labels.
- **Places** [51] is another scene dataset with similar concept coverage as SUN. A chosen subset of 10,000 images across 50 classes (not contained in ImageNet-1k) are used.
- **Textures** [4] contains 5,640 real-world texture images under 47 categories. We use the entire dataset for evaluation.

**CIFAR benchmark**    CIFAR-10 and CIFAR-100 [22] are widely used as ID datasets in the literature, which contain 10 and 100 classes, respectively. We use the standard split with 50,000 training images and 10,000 test images. We evaluate our approach on four common OOD datasets, which are listed below:

- **SVHN** [33] contains color images of house numbers. There are ten classes of digits 0-9. We use the entire test set containing 26,032 images.

- **LSUN** [50] contains 10,000 testing images across 10 different scenes. Image patches of size $32{\times}32$ are randomly cropped from this dataset.
- **Places365** [51] contains large-scale photographs of scenes with 365 scene categories. There are 900 images per category in the test set. We randomly sample 10,000 images from the test set for evaluation.
- **Textures** [4] contains 5,640 real-world texture images under 47 categories. We use the entire dataset for evaluation.

## B.2 Details of Baselines

For the reader's convenience, we summarize in detail a few common techniques for defining OOD scores that measure the degree of ID-ness on the given sample. By convention, a higher (lower) score is indicative of being in-distribution (out-of-distribution).

**MSP** [13] propose to use the maximum softmax score to detect OOD samples.

**ODIN** [27]  Liang et al. improved OOD detection with temperature scaling and input perturbation. In all experiments, we set the temperature scaling parameter $T = 1000$. Note that this is different from calibration, where a much milder $T$ will be employed. While calibration focuses on representing the true correctness likelihood of in-distribution data, the OOD scores proposed by ODIN are designed to maximize the gap between ID and OOD data and may no longer be meaningful from a predictive confidence standpoint. For ImageNet, we found the input perturbation does not further improve the OOD detection performance and hence we set $\epsilon = 0$. Following the setting in [27], we set $\epsilon$ to be 0.004 for CIFAR-10 and CIFAR-100.

**Energy** [29] Liu et al. first proposed using energy score for OOD uncertainty estimation. The energy function maps the logit outputs to a scalar $S_{\text{Energy}}(\mathbf{x}; f) \in \mathbb{R}$, which is relatively lower for ID data:

$$S_{\text{Energy}}(\mathbf{x}; f) = -\log \sum_{i=1}^{C} \exp(f_i(\mathbf{x})). \tag{10}$$

Note that Liu et al. [29] used the *negative energy score* for OOD detection, in order to align with the convention that $S(\mathbf{x}; f)$ is higher (lower) for ID (OOD) data. Energy score is a hyperparameter-free score.

**Mahalanobis** [26]  Lee et al. use multivariate Gaussian distributions to model class-conditional distributions of softmax neural classifiers and use Mahalanobis distance-based scores for OOD detection. We use 500 examples randomly selected from ID datasets and an auxiliary tuning dataset to train the logistic regression model and tune the perturbation magnitude $\epsilon$. The tuning dataset consists of adversarial examples generated by FGSM [10] with a perturbation size of 0.05. The selected $\epsilon$'s are 0.001, 0.01, and 0.005 for ImageNet-1k, CIFAR-10, and CIFAR-100, respectively.

## B.3 Software and Hardware

**Software**  We run all experiments with Python 3.8.0 and PyTorch 1.6.0.

**Hardware**  All experiments are run on NVIDIA GeForce RTX 2080Ti.

## C Complete Results Under Different $L_p$-norms

In Table 7, we report the OOD detection performance of using more $L_p$-norms as OOD scores, in addition to the 6 norms shown in Figure 4. $L_1$-norm achieves the best overall performance.

| OOD Data | Norm | FPR95 ↓ | AUROC ↑ |
|---|---|---|---|
| iNaturalist | $L_{0.3}$ | 60.30 | 83.45 |
| | $L_{0.5}$ | 65.28 | 81.44 |
| | $L_{0.8}$ | 66.25 | 82.69 |
| | $L_1$ | **50.03** | **90.33** |
| | $L_2$ | 65.88 | 87.68 |
| | $L_3$ | 71.15 | 85.71 |
| | $L_4$ | 73.53 | 84.67 |
| | $L_5$ | 74.66 | 84.13 |
| | $L_6$ | 75.19 | 83.83 |
| | $L_\infty$ | 76.42 | 83.18 |
| SUN | $L_{0.3}$ | 48.53 | 86.73 |
| | $L_{0.5}$ | 51.98 | 85.13 |
| | $L_{0.8}$ | 54.52 | 84.75 |
| | $L_1$ | **46.48** | **89.03** |
| | $L_2$ | 76.09 | 80.30 |
| | $L_3$ | 80.87 | 77.85 |
| | $L_4$ | 82.59 | 76.65 |
| | $L_5$ | 83.37 | 76.01 |
| | $L_6$ | 83.73 | 75.64 |
| | $L_\infty$ | 84.41 | 74.83 |
| Places | $L_{0.3}$ | 64.10 | 81.46 |
| | $L_{0.5}$ | 67.39 | 79.79 |
| | $L_{0.8}$ | 69.01 | 79.83 |
| | $L_1$ | **60.86** | **84.82** |
| | $L_2$ | 80.14 | 77.80 |
| | $L_3$ | 82.68 | 75.69 |
| | $L_4$ | 83.70 | 74.61 |
| | $L_5$ | 84.24 | 74.03 |
| | $L_6$ | 84.34 | 73.70 |
| | $L_\infty$ | 85.01 | 72.95 |
| Textures | $L_{0.3}$ | 53.21 | 82.82 |
| | $L_{0.5}$ | 41.54 | 88.46 |
| | $L_{0.8}$ | **33.40** | **92.07** |
| | $L_1$ | 61.42 | 81.07 |
| | $L_2$ | 85.20 | 70.21 |
| | $L_3$ | 87.50 | 67.54 |
| | $L_4$ | 88.30 | 66.68 |
| | $L_5$ | 88.49 | 66.40 |
| | $L_6$ | 88.37 | 66.29 |
| | $L_\infty$ | 88.40 | 66.13 |
| Average | $L_{0.3}$ | 56.54 | 83.62 |
| | $L_{0.5}$ | 56.55 | 83.71 |
| | $L_{0.8}$ | 55.80 | 84.84 |
| | $L_1$ | **54.70** | **86.31** |
| | $L_2$ | 76.83 | 79.00 |
| | $L_3$ | 80.55 | 76.70 |
| | $L_4$ | 82.03 | 75.65 |
| | $L_5$ | 82.69 | 75.14 |
| | $L_6$ | 82.91 | 74.87 |
| | $L_\infty$ | 83.56 | 74.27 |

Table 7: OOD detection performance comparison under different $L_p$-norms. We use a ResNetv2-101 architecture pre-trained on ImageNet-1k.

## D    Complete Results Under Different Scaling Temperatures

In addition to Figure 5, we report the OOD detection performance under more scaling temperatures in Table 8. $T = 1$ achieves the best average performance.

| Temperature | iNaturalist FPR95 ↓ | iNaturalist AUROC ↑ | SUN FPR95 ↓ | SUN AUROC ↑ | Places FPR95 ↓ | Places AUROC ↑ | Textures FPR95 ↓ | Textures AUROC ↑ | Average FPR95 ↓ | Average AUROC ↑ |
|---|---|---|---|---|---|---|---|---|---|---|
| 0.0625 | 77.70 | 74.46 | 61.87 | 78.80 | 76.42 | 72.84 | 67.80 | 72.84 | 70.95 | 74.74 |
| 0.125 | 77.47 | 74.68 | 61.84 | 78.90 | 76.35 | 72.97 | 67.73 | 72.93 | 70.85 | 74.87 |
| 0.25 | 76.64 | 75.55 | 61.52 | 79.25 | 75.95 | 73.48 | 67.41 | 73.30 | 70.38 | 75.40 |
| 0.5 | 70.80 | 80.01 | 58.83 | 81.47 | 72.95 | 76.38 | 65.30 | 75.29 | 66.97 | 78.29 |
| 1 | **50.03** | **90.33** | 46.48 | 89.03 | 60.86 | 84.82 | **61.42** | **81.07** | **54.70** | **86.31** |
| 2 | 58.06 | 87.33 | **41.03** | **91.67** | **55.71** | **87.54** | 65.94 | 77.98 | 55.19 | 86.13 |
| 4 | 71.69 | 79.68 | 47.37 | 89.87 | 61.96 | 85.97 | 71.21 | 70.96 | 63.06 | 81.62 |
| 8 | 77.47 | 76.01 | 52.85 | 87.87 | 66.76 | 83.82 | 73.62 | 67.50 | 67.68 | 78.80 |
| 16 | 79.54 | 74.71 | 55.08 | 86.93 | 68.69 | 82.80 | 74.73 | 66.24 | 69.51 | 77.67 |
| 32 | 80.25 | 74.18 | 56.22 | 86.49 | 69.81 | 82.33 | 75.02 | 65.71 | 70.33 | 77.18 |
| 64 | 80.53 | 73.94 | 56.53 | 86.28 | 70.08 | 82.11 | 75.09 | 65.48 | 70.56 | 76.95 |
| 128 | 80.70 | 73.83 | 56.78 | 86.18 | 70.24 | 82.00 | 75.11 | 65.36 | 70.71 | 76.84 |
| 256 | 80.81 | 73.77 | 56.94 | 86.13 | 70.32 | 81.94 | 75.12 | 65.31 | 70.80 | 76.79 |
| 512 | 80.84 | 73.74 | 57.01 | 86.10 | 70.39 | 81.91 | 75.16 | 65.28 | 70.85 | 76.76 |
| 1024 | 80.85 | 73.73 | 57.05 | 86.09 | 70.41 | 81.90 | 75.16 | 65.27 | 70.87 | 76.75 |

Table 8: OOD detection performance comparison under different temperatures. We use a ResNetv2-101 architecture pre-trained on ImageNet-1k.

## E    Comparison with Lee and AlRegib

Lee and AlRegib [25] proposed an OOD detection framework using the $L_2$-norm of gradients. Importantly, they do not directly use gradient norms for OOD detection, but instead, use them as the input for training a separate binary classifier. The binary classifier is trained on layer-wise gradients from both ID and OOD data. Since `GradNorm` does not require any OOD data, these two methods are not directly comparable. For a fair comparison, we use the gradients of uniform noise as a surrogate of OOD data to train the binary classifier. Following the original work, we use a 40%-40%-20% train-validation-test split. Results are shown in Table 9. `GradNorm` outperforms [25] by 15.45% in FPR95 on average.

| Method | iNaturalist FPR95 ↓ | iNaturalist AUROC ↑ | SUN FPR95 ↓ | SUN AUROC ↑ | Places FPR95 ↓ | Places AUROC ↑ | Textures FPR95 ↓ | Textures AUROC ↑ | Average FPR95 ↓ | Average AUROC ↑ |
|---|---|---|---|---|---|---|---|---|---|---|
| Lee and AlRegib [25] | 75.49 | 72.30 | 73.82 | 82.61 | 87.90 | 74.00 | **43.39** | **84.16** | 70.15 | 78.27 |
| **GradNorm (ours)** | **50.03** | **90.33** | **46.48** | **89.03** | **60.86** | **84.82** | 61.42 | 81.07 | **54.70** | **86.31** |

Table 9: Comparison of `GradNorm` Lee and AlRegib's approach on ImageNet benchmark. The classification model is the same as in Table 1 (standard ResNetv2-101 model pre-trained on ImageNet). For Lee and AlRegib's method, we use the gradients of uniform noise as a surrogate of OOD data to train the binary classifier.