# OpenReview forum: "On the Importance of Gradients for Detecting Distributional Shifts in the Wild"
_NeurIPS.cc/2021/Conference — NeurIPS 2021 Poster_

### Official Review · Reviewer_7r4h · 2021-07-14

**Rating:** 6
**Confidence:** 3

**Summary:**

This work presents a new method for detecting out-of-distribution inputs that relies on the gradient norm of the KL divergence between the model’s softmax output and a uniform probability distribution.  The method is easy to implement (using the norm of the gradient of a standard cross entropy loss with a uniform label vector), can be used at inference time without access to true labels, and does not require training an additional model.


The authors present empirical evidence that their method outperforms prior work by a significant margin, reducing the false positive rate (FPR95) for out-of-distribution detection by 10.89% and 14.47% on ImageNet and CIFAR-10 respectively. Additionally, the authors perform an ablation study that explores how their method performs as they vary layer parameters, type of target distribution used (uniform versus real), temperature, type of gradient norm, and using only individual terms that form a decomposition of the gradient norm.


**Limitations And Societal Impact:**

The author should discuss potential negative societal impacts of their work in the main paper.

**Main Review:**

**Major Comments**

1. *Evaluation on covariate shifts.* The authors only evaluate on datasets where the class hierarchy is distinct from that of the in-distribution dataset.  It is not clear that GradNorm would still perform well on covariate shifts. The authors claim that the GradNorm technique is effective since the in-distribution predictions tend to concentrate on the ground truth class and are thus distributed less uniformly.  It’s possible that covariate-shifted data would similarly have a higher GradNorm because the predictions would concentrate on the ground truth class. For ImageNet, the authors could have evaluated on ImageNetV2, ObjectNet, ImageNet-R, ImageNet-C, ImageNet-Vid-Robust, or YouTube-Bounding-Boxes. For CIFAR, the authors could have evaluated on CINIC-10, CIFAR-10.1, or CIFAR-10C.

2. *Hyperparameter to set the gradient norm threshold.*  The authors claim that their work is hyperparameter free if one sets the temperature to T=1.  However, isn’t the threshold on the gradient norm for separating in-distribution versus out-of-distribution data ($\gamma$) an important hyperparameter?  Clarifying how this parameter is set for different datasets would improve the paper.

**Minor comments**

3. *Calibration*. From Figure 7, it looks like the authors only tried one temperature value that was smaller than T=1. The authors also do not give an explanation for why temperatures smaller than T=1 would work. It would also be interesting to see if the optimal temperature for calibration would fare any better than T=1.


4. *Explanation of prior work.* It would be helpful if the authors gave a brief description of prior work they compare against, e.g. MSP, ODIN, Generalized ODIN, Energy or Mahalanobis.

5. *Effect of label noise.* Mislabeled images might force the in-distribution predictions to be more uniformly distributed. Have the authors looked at whether GradNorm can be used to detect label noise? Would the technique be able to distinguish between out of distribution and label noise?

6. *Organization.* The section “Effect of temperature scaling” is an ablation study and should be included in Section 4.


**Originality**: The method is novel and the authors describe how it is distinct from prior work.

**Quality**: See comments above.

**Clarity**: Paper is clearly written, but could be improved by clearer descriptions of related work.

**Significance**: Problem is important, and proposed algorithm is easy to implement and deploy.

If major comments are addressed, I would consider raising my score to a 6.

**Time Spent Reviewing:**

3.5

---

> ### Author Response · Authors · 2021-08-10
> **Thank you for the constructive feedback**
>
> We appreciate the reviewer's detailed review and comments, which raise insightful points on not only improving our draft but also extending our work in future research.
>
> We address the reviewer’s comments below:
>
> > **Q1. Covariate shifts**
>
> We are glad you brought it up! Literature in OOD detection is commonly concerned about model reliability and **detection** of shifts where the OOD inputs have disjoint labels and therefore _should not be predicted by the model_. Covariate shifts (such as CIFAR-10C) are less common for OOD detection evaluation, and are more relevant for evaluating model robustness and **generalization** performance &mdash; in which case the goal is to make the model classify accurately into the ID classes (see evaluation and metrics in [Hendrycks and Dietterich 2019] for example). As the reviewer said, covariate-shifted data is therefore expected to have a higher GradNorm because we want the predictions to concentrate on the ground truth class (if the model indeed can generalize well). As suggested, we have indeed verified this on pairs (e.g., CIFAR-10 vs. CIFAR-10C) and observed that the GradNorm distribution of CIFAR-10C is similar to that of CIFAR-10. We will acknowledge this point in our discussion.
>
> > **Q2. Hyperparameter to set the gradient norm threshold**
>
> Fair point! In OOD literature, $\gamma$ is often chosen based on a threshold where 95% of ID data is classified correctly (i.e., above the threshold). Therefore, $\gamma$ is typically not considered as a hyperparameter for a given percentile (e.g., 95%). Strictly speaking, it can be a hyperparameter without the specified percentile. For each ID dataset, we can calculate the GradNorm for the ID test data, and then choose the 95% cutoff as $\gamma$\. Furthermore, when we evaluate AUROC, $\gamma$ is not required since the metric is threshold-independent. We have added clarification on this in our updated draft. Thanks for pointing that out!
>
> > **Q3. Effect of temperature**
>
> We actually explored several different temperatures (T=0.0625, 0.125, 0.25, 0.5) with T<1. They all perform very similarly and none of them outperform T=1. Therefore, we only included T=0.5 in Fig 7 for visual clarity. For completeness, we have included numerical results on this in Appendix (also see attached below). Results are averaged over 4 OOD datasets.
>
> | Temperature | 0.0625 | 0.125 |  0.25 |  0.5  |   1   |   2   |   4   |   8   |   16  |   32  |   64  |  128  |  256  |  512  |  1024 |
> |-------------|:------:|:-----:|:-----:|:-----:|:-----:|:-----:|:-----:|:-----:|:-----:|:-----:|:-----:|:-----:|:-----:|:-----:|:-----:|
> |    AUROC    |  74.74 | 74.87 | 75.40 | 78.29 | **86.31** | 86.13 | 81.62 | 78.80 | 77.67 | 77.18 | 76.95 | 76.84 | 76.79 | 76.76 | 76.75 |
> |    FPR95    |  70.95 | 70.85 | 70.38 | 66.97 | **54.70** | 55.19 | 63.06 | 67.68 | 69.51 | 70.33 | 70.56 | 70.71 | 70.80 | 70.85 | 70.87 |
>
>
> > **Q4. Explanation of prior work**
>
> Great suggestion! In **Appendix A.2**, we have a section of details on how we reproduce these baseline methods. We have extended this section to include a brief introduction of each baseline method.
>
> > **Q5. Effect of label noise**
>
> This is a very interesting point and we have investigated further. Let’s say our training data is defined on a joint distribution $\mathcal{X}\times \mathcal{Y}$, and the in-distribution $P$ is the marginal distribution over the label space. Let’s consider two types of “mislabeled images” &mdash; which can either come from a different distribution, or the same marginal distribution $P$ but just with label swapped.
> + If the reviewer is referring to the former type, then our method can detect those as out-of-distribution but not distinguish the label noise/correctness. This is because GradNorm by design is label-agnostic and only tries to differentiate in the image space (see definition in L74-78).
> + If the image itself is sampled from the same marginal distribution $P$, our current method GradNorm cannot detect these because they are still considered ID data in the image space. These images (regardless of the label) can still lead to a less uniformly distributed prediction. Say a “cat” is mislabeled as a “dog” in CIFAR-10, the prediction in testing can still be concentrated on the “cat” class.
> We believe designing OOD detection methods tailored for label noise can be a really cool direction that is worth future research!
>
> > **Q6. Organization**
>
> As suggested, we will move “Effect of temperature scaling” into Section 4.2.
>
> > **Q7. Potential negative impact**
>
> We have revised our draft to reflect this missing point. In particular, we do not foresee the negative impact of our work. Our work can potentially lead to broad impact and benefit safety-critical applications where OOD data could emerge, such as in healthcare and also in autonomous driving. Thank you again for the suggestions!

---

> > ### Comment · Reviewer_7r4h · 2021-08-25
> > **Response**
> >
> > Thank you for the response and the clarifications of all my questions.
> >
> > For the Major Concerns, I'm satisfied with the response to Q2. For Q1, my understanding now is that GradNorm would not be effective in detecting covariate shifts. That said, after discussion with other reviewers, we agree that the setting where the class label sets are distinct is still an important setting to consider. So, I am updating my score to 6.

---

> > > ### Comment · Reviewer_7r4h · 2021-08-25
> > > **one ask**
> > >
> > > Though I will ask the reviewers to please clarify this as a limitation in the paper!

---

> > > > ### Author Response · Authors · 2021-08-25
> > > > **Thank you for the followup**
> > > >
> > > > Thank you! We are glad that the clarifications were helpful. Indeed, we have already included a discussion on the limitation as suggested (see summary of changes post). We believe the discussion would be beneficial for the research community.

---

### Official Review · Reviewer_3Auv · 2021-07-15

**Rating:** 8
**Confidence:** 4

**Summary:**

The work proposes a novel method for OOD detection. The method itself uses as scoring function the norm of the gradient of the KL divergence of the softmax vector to a uniform distribution. Importantly, this method does not require ground-truth OOD data and does not require hyperparameter tuning. The main contribution of the work is empirical demonstration that the method achieves state-of-the-art performance on a variety of datasets, as well as ablations to further understanding into the effectiveness of the proposed method.


**Limitations And Societal Impact:**

copied from above:

There was no discussion of limitations or negative potential impact, which is a problem considering the explicit NeurIPS requirement this year. As a suggestion to the authors, I think the derivation of the gradnorm decomposition in section 5 could be left to the appendix so that more room is left in the main body for a discussion of limitations and negative potential impact of the work.


**Main Review:**

I think this paper is overall a solid contribution and would recommend acceptance for the following reasons.
- The method is relatively simple and practical. The only operation is taking the norm of the gradient of the KL divergence, and does not require ground-truth label information. This latter fact is especially important as it might obviate the need for excessive human intervention in practice. Methods that use ground-truth label information almost seem to be “cheating”, and seem to have a scaling problem by requiring samples from the OOD dataset to begin with.
- The writing and organization were exceptionally clear throughout. I do not actively do work in OOD detection, yet I found the work easy to follow.
- The experimental methodology is high-quality and demonstrates a significant improvement over baselines.
    - The baselines were tuned, which is especially important because the paper has many experiments on imagenet, but some of the baselines were originally only tested on the cifar datasets.
    - There is a large variety of datasets considered as well, which helps to give a balanced sense of the performance of the proposed method. Indeed, the method performs worse than some baselines on some datasets, yet performs the best on average across all datasets.
    - The ablations (to KL divergence, different L_p norms, different temperatures, gradnorm of different layers, one-hot vs. all labels, feature and output space components) were extensive and interesting. I especially found comparison to using just the norm of the KL divergence interesting; based on the motivation on line 35, I would have expected KL divergence to do better than the proposed method (more on this point below).
- I liked the mathematical analysis in section 5 to decompose the gradnorm into a feature space quantity and an output space quantity, and especially appreciated how it led to another ablation experiment. Given that the main contribution of the paper is empirical, I saw this section as a nice bonus to further understanding of the proposed method, and in future work would look forward to seeing more extensive mathematical analysis.

There did remain limitations of the paper, but these were not enough to outweigh its advantages above. I think addressing these limitations would make the paper stronger.
- I found the intuition for the gradnorm to be lacking. The main intuition is on lines 35 and 94, where I get the impression the ID data should have a higher KL divergence. However, it’s not clear to me that the gradient norms for ID data should be higher. The experiments certainly demonstrate that the gradient norm is a good discriminator of ID and OOD data, but I would have liked better intuition for this result.
- There was no discussion of limitations or negative potential impact, which is a problem considering the explicit NeurIPS requirement this year. As a suggestion to the authors, I think the derivation of the gradnorm decomposition in section 5 could be left to the appendix so that more room is left in the main body for a discussion of limitations and negative potential impact of the work.


**Time Spent Reviewing:**

3

---

> ### Author Response · Authors · 2021-08-10
> **Thank you for the encouraging and constructive feedback**
>
> We are more than encouraged that the reviewer found the paper a solid contribution and exceptionally clear. We particularly appreciate the reviewer for the constructive feedback. We address the reviewer’s comments below:
>
> > **Q1. Intuition on ID data having higher KL divergence**
>
> Excellent question! We have provided analysis and intuition on this in **L187-196**. In short, whether the gradient norm of ID is higher depends largely on the target function (one-hot vs. uniform target). Specifically, using uniform target (ours), gradients of ID data have larger magnitudes than those of OOD data, as the softmax prediction tends to be less uniformly distributed (and therefore results in larger KL divergence). In contrast, the gradient norm using one-hot targets shows the opposite trend, with ID data having lower magnitudes. This is also expected since the training objective explicitly minimizes the cross-entropy loss, which results in smaller gradients for the majority of ID data. We have revised our introduction to better highlight this.
>
> > **Q2. Limitations and potential negative impact**
>
> We like the suggestion of moving some derivation in Section 5 and add discussion on limitations and societal impact. We have already revised our draft to reflect this. In particular, we do not foresee the negative impact of our work. Our work can potentially lead to broad impact and benefit safety-critical applications where OOD data could emerge, such as in healthcare and also in autonomous driving. One limitation is that our study is primarily evaluated on image datasets. Although this is more common in OOD detection literature, we are interested in expanding the future study to other data modalities.

---

> > ### Comment · Reviewer_3Auv · 2021-08-14
> > **thanks for the explanation!**
> >
> > I think I understand the intuition now. Thanks for the explanation!

---

### Official Review · Reviewer_p1oz · 2021-07-15

**Rating:** 7
**Confidence:** 4

**Summary:**

The work presents GradNorm which uses the norm of gradient, back-propagated from the KL divergence between the softmax output and a uniform probability distribution, as the uncertainty score for OOD detection. The experimental results show that the proposed method outperforms the baselines on ImageNet1k vs OOD and CIFAR vs OOD benchmarks. Ablation studies are presented to help better understand the proposed method. The paper is written well and the experiments are comprehensive .


**Limitations And Societal Impact:**

The limitations were not addressed. It is unknown if the method works well on discrete data like text. That can be left as a future work.

Societal Impact was not discussed.

**Main Review:**

- Originality: The proposed method GradNorm is novel, simple and efficient, as demonstrated by the experimental results. Most existing work rely on the output (MSP, ODIN, SNGP) or feature space (Mahalanobis) for deriving OOD scores. This work suggests making use of the information from the gradient space for OOD detection.

- Quality: The results are presented clearly. Comprehensive ablation studies were conducted and that did help me better understand the method.  It would be great to include the benchmark dataset CIFAR10 vs CIFAR100 as one of the near-OOD tasks.

- Clarity: The work is well written and the content is well organized. One comment I have is regarding one of the baseline methods, called Lee and AlRegib [28]. It is not clear how the experiment for this method was conducted exactly. As mentioned in line 151-152, Lee and AlRegib requires “a binary classifier trained on the OOD datasets, which can unfairly overfit to the test data and does not suit OOD-agnostic settings in the real world”. But it is also stated in line 139-140 that “for a fair comparison, all the methods use the same pre-trained backbone, without regularizing with auxiliary outlier data”.

- Significance: the results are important. This work inspires the future work to use the information from the gradient space for OOD detection. Combining the information from gradient space with that from the output space and feature space, would potentially result in a better OOD score.


Update: Thank the authors for responding to my questions. I keep my score. I would suggest the authors to incorporate the reviewers' comments to the final version.

**Time Spent Reviewing:**

3

---

> ### Author Response · Authors · 2021-08-10
> **Thank you for the constructive feedback**
>
> We are encouraged that the reviewer found our method novel, simple, and efficient. We thank the reviewer for the helpful comments and suggestions, which we address below:
>
> > **Q1. Clarification on reproducing [1]**
>
> To clarify, our statement in L139-140 meant to describe the pre-training of neural network backbone is purely based on optimizing the classification accuracy of ID data, without using an outlier exposure type of approach during model training. All the methods (including ours), extract information (e.g., features, outputs, or gradients) from such the same network backbone. The only exception is [1], which used a 40%-40%-20% train-validation-test split of the OOD dataset to train a separate binary classifier, which can unfairly overfit to the test data and is a known red flag in OOD detection research. Literature on OOD detection has used surrogate noise to circumvent this issue. For a fair comparison, we follow a similar practice using gradients of random noise to train the binary classifier. We have clarified this and added details in our updated draft.
>
> > **Q2. CIFAR-10 vs CIFAR-100 experiment**
>
> Thanks for the suggestion! We agree that this will be an interesting experiment to add. We will make sure to include results in our final draft.
>
> > **Q3. Limitations and potential negative impact**
>
> We absolutely agree that OOD detection on text is an interesting topic, and we plan to explore how GradNorm performs in this context in future work! Our work can potentially lead to broad impact and benefit safety-critical applications where OOD data could emerge, such as in healthcare and also in autonomous driving. We have expanded our discussion on this. Thank you again for pointing this out.
>
>
> [1] Jinsol Lee and Ghassan AlRegib. Gradients as a measure of uncertainty in neural networks. ICIP. 2020

---

### Official Review · Reviewer_UMiA · 2021-07-16

**Rating:** 6
**Confidence:** 3

**Summary:**

The manuscript proposes using the gradient norm of a trained classifier for out-of-distribution detection. Concretely, the norm of the gradient of the KL divergence between the actual prediction and the uniform prediction with regard to the network parameters is used to distinguish in-distribution and out-of-distribution data. Results on several benchmarks show this method has better OOD detection performance than some other methods.


**Limitations And Societal Impact:**

I only saw a very minimal discussion of this at the beginning of the introduction.


**Main Review:**

** Update**: Increase score to 6 due to added experiment SVHN vs CIFAR10

I found the manuscript fairly easy to read and understand, the method seems sensible and the results are interesting.

I see some weaknesses in the discussion of related work and in the evaluation.

I would suggest to more clearly and extensively discuss in one place exactly the relation of this work to [28]. And refer to that more extensive discussion in the introduction already, as it seems very crucial in order to determine what is novel about this work. Like it seems to me that that [28] use may be the same gradients as in this manuscript? From their “To generate gradients, we utilize a confounding label where all classes are positive (i.e. a vector of all 1’s).” Or is there a difference still between the two? Somehow it seems the motivation in this manuscript and theirs is opposite, judging from ““If the model is unfamiliar with the given input, however, the amount of updates required to 1) learn new features to properly represent the new input, and 2) associate the new features with the confounding label, would be larger” [28] and “Our operating hypothesis is that using the KL divergence for backpropagation, gradient norm is higher for ID data than that for OOD data“ (this manuscript). Do you have any comments on that? Or maybe I have some misunderstanding?

Regarding empirical evaluation, where do the numbers for [28] come from? Comparing Table 4 in this manuscript to Table 1 in [28] the numbers seem completely different. Also, shouldn’t your method perform worse since their method even trains an extra classifier that sees OOD data? It would be good to also clearly show what is the difference in method that causes your results to be better, if they are indeed so.

Also I think it would be good if you could clearly delineate what methods you are including and excluding in your comparison. There are so many variants for OOD detection methods, generative, discriminative, with outliers, without outlines etc. and it would be good to know what you consider fair comparisons and what not.

It would also be important to report values not only for CIFAR10 in-dist vs SVHN OOD but also reverse case, SVHN in-dist vs CIFAR10 OOD as some methods degrade the performance in the latter “easier” setting.

How do the results in 4.2 table 3 relate to the results in table 2? Seems even the best results from table 3 do not reach the results from table 2?

You should include https://arxiv.org/abs/1911.07421 and https://arxiv.org/abs/2006.10848 in related work generative models and your supplementary comparison for generative models. https://arxiv.org/abs/2007.05566 could be another interesting related work.

Analysis and Ablation studies seem interesting to me.

What was the motivation for training your own networks and not using pretrained networks on CIFAR10 etc.?



**Time Spent Reviewing:**

3

---

> ### Author Response · Authors · 2021-08-10
> **Thank you for the constructive feedback**
>
> We are glad the reviewer found that the paper is easy to understand, the method is sensible, and the results are interesting. We appreciate the detailed comments and suggestions, which have helped us improve our draft.
>
> We address the comments raised by the reviewer below:
>
> > **Q1. Difference w.r.t [1]**
>
> We absolutely agree on the importance of discussing this! In fact, we already summarized the differences (see **L148-157** in Section 4.1 and in introduction). As suggested, we also highlighted this in the updated draft. More details are elaborated below:
>
> + First, the way we utilize gradients for OOD detection is very different from [1]. [1] did not directly use gradient norms, but instead, used them as the input for training a separate binary classifier. The training of the binary classifier in [1] is problematic as it directly uses the OOD test data. Literature in OOD detection generally does not assume the knowledge of test OOD data, otherwise one can unfairly overfit to the test data. In contrast, our method mitigates the shortcomings in that GradNorm (1) does not require any separate model training, (2) is hyperparameter-free, and (3) is suitable for OOD-agnostic settings.
> + Second, as several reviewers recognized, we provide comprehensive ablation studies and analyses on different design choices in using gradient-based methods for OOD detection (network architectures, gradients at different layers, loss functions for backpropagation, different $L_p$-norms, diverse in-distribution and OOD datasets, and different temperatures, etc.), which were previously not studied in [1]. We believe such a thorough understanding will be valuable for the research community. In particular, [1] utilizes $L_2$-norm gradients without comparing them with other norms. Our ablation study leads to the new finding that $L_1$-norm works best among all $L_p$ variants, and outperforms $L_2$-norm by up to 22.31% in FPR95. Furthermore, [1] utilizes gradients from _all layers_ to train a separate binary classifier which can cause the computational cost to become intractable for deeper and larger models. In contrast, with GradNorm we show that the _last layer_ gradient will always yield the best performance among all gradient set selections. Consequently GradNorm will incur negligible computational cost in comparison to [1].
> + Although both methods use a uniform vector as the target, we have provided sound technical reasoning and analysis. Our study mitigated technical flaws in [1], particularly including the paradox of seemingly opposite motivation as the reviewer raised. In [1], the authors argue that gradients of ID data should be smaller than those of OOD data. However, as far as we know, no direct evidence was provided to support this argument. Even though [1] provided an illustrative example in Fig. 3(a) demonstrating gradients of ID data are lower than OOD data, to the best of our knowledge, the figure is based on the “per-class average” and the one-hot target was used. We have carefully examined this and showed that trends are the opposite between using a uniform target and a one-hot target for backpropagation (see Figure 2 in our paper). We have verified that our motivation and hypothesis &mdash; gradients for ID data should be larger than those for OOD data &mdash; are indeed correct via comprehensive experiments.
> + Lastly, our theoretical analysis in Section 5 and method interpretation from the KL divergence perspective are also completely new and not presented in [1].
>
> > **Q2. Clarification on the performance of [1]**
>
> The numbers in [1] and ours are not a fair comparison, due to the data assumption discrepancy (uses OOD test data vs. OOD-agnostic). In particular, [1] used a 40%-40%-20% train-validation-test split of the OOD dataset to train the binary classifier, which can unfairly overfit to the test data and is a known red flag in OOD detection research. Literature on OOD detection more often used surrogate noise to circumvent this issue. For a fair comparison, we followed a similar practice and used gradients of random noise to train the binary classifier. As discussed above, our method also shows a strong advantage due to the $L_1$-norm instead of the $L_2$-norm as in [1]. We have clarified this in our updated draft.
>
> > **Q3. Discussion on fair comparison**
>
> We completely agree on the importance of specifying the ground for a fair comparison. As stated on **L139** and Section 6, for a fair comparison, we primarily compare with methods utilizing a pre-trained discriminative network without regularizing with auxiliary outlier data. We do not consider the outlier-based approach as it imposes a strong data assumption, and it can be prohibitive to construct an auxiliary outlier dataset for the in-distribution dataset ImageNet. Our main results presented in Tables 1, 2 & 4 are based on the discriminative family, which are all comparable in a fair setting.
>
> > **Q4. Experiment on SVHN in-dist vs CIFAR-10 OOD**
>
> We observe an AUROC of 95.43% and an FPR95 of 20.58% in the reversed setting suggested. The training configuration is the same as described in Section 4.3. In comparison, the CIFAR-10 in-dist vs. SVHN OOD has an AUROC of 96.66% and an FPR95 of 17.76%. We didn’t observe significant degradation in the reversed setting.
>
> > **Q5. Clarification on Table 2 and Table 3**
>
> Thanks for the observation! Table 3 should be read together with Table 1, since the network we used in Section 4.2 is ResNetv2-101 instead of DenseNet-121. The best result in Table 3 is exactly what is shown as GradNorm in Table 1. We have added some clarification in Section 4.2 in the draft.
>
> > **Q6. Extra related work**
>
> Thanks very much for pointing this out! We have further added these papers into our related work section!
>
> > **Q7. Training your own networks vs using pre-trained networks**
>
> Our method works on standard pre-trained networks. We trained from scratch to ensure we know the exact training configurations of training to facilitate reproducibility and documentation.
>
> > **Q8. Impact statement**
>
> Our work does not foresee a negative impact on society, and we have revised to include a discussion on this.
>
>
>
> [1] Jinsol Lee and Ghassan AlRegib. Gradients as a measure of uncertainty in neural networks. ICIP. 2020

---

> > ### Comment · Reviewer_UMiA · 2021-08-25
> > **Thanks for added experiments**
> >
> > Thanks for the answers.
> > I agree it is not fair to compare methods that access OOD samples to those that do not. Still, it should be made clear what exactly you are comparing to, including that you are not comparing to reported numbers, but to some reimplementation of yours. These details should be clearly referenced from the table caption as well.
> > Thanks for performing the SVHN vs CIFAR10 experiment. Regarding SVHN in-dist, ~95.43% is a degradation to *simple baselines* to me. For example, if you train a well-performing generative model with maximum likelihood on SVHN and use raw likelihoods as OOD scores, you can get 99%+ AUC for SVHN vs CIFAR10.  So I think these results should also be part of your manuscript, at the least in the supplementary with a reference to that supplementary part in the main manuscript. That will greatly help future comparisons.
> >
> > Due to the added experiment, I increase my score to 6.

---

> > > ### Author Response · Authors · 2021-08-25
> > > **Thank you for the followup**
> > >
> > > We appreciate the updated score and additional feedback, which helped us strengthen the manuscript. We have incorporated the suggested changes in our revised draft. Thank you again!

---

### Author Response · Authors · 2021-08-10
**Summary of response -- thanks to all reviewers for thorough and insightful feedback**

We are encouraged to see that reviewers find that our method **novel**, **simple** and **a solid contribution** (*R2*, *R3*, *R4*), and the results and analysis are **comprehensive** (*R2*, *R3*), **interesting** (*R1*, *R3*) and **high-quality** (*R3*), with **significant improvements** over baseline methods (*R2*, *R3*, *R4*). We are equally glad that all reviewers found the paper **easy to read** and **exceptionally clear** (*R1*, *R2*, *R3*, *R4*). We appreciate that *R3* thinks our work is **practical and scalable** because it obviates the need for excessive human intervention or OOD data to begin with.

We have addressed the reviewers’ comments and concerns in individual responses to each reviewer. The reviews allowed us to improve our draft and the changes made in the revised draft are summarized below. We will incorporate remaining suggestions in our final draft.

+ [*R1*, *R2*, *R3*, *R4*] Updated Section 1 and Section 7 to include discussion of limits and potential negative impacts.
+ [*R1*, *R2*] Updated Section 4.1 to clarify the reproduction and performance of [1].
+ [*R1*] Expanded L148-157 in Section 4.1 discussing differenences w.r.t [1].
+ [*R1*] Added clarification in Section 4.2 about the network we use.
+ [*R1*] Added extra related work in Section 6.
+ [*R3*] Revised our introduction to better highlight our intuition from L187-196 &mdash; whether ID data has higher KL divergence largely depends on the target vector (one-hot vs. uniform).
+ [*R4*] Added clarification in Section 4 about how to determine $\gamma$, and discussion on covariate shift.
+ [*R4*] Added more results of temperatures with $T<1$ in Appendix.
+ [*R4*] Expanded Appendix A.2 on the baseline methods.


\* For brevity, we refer to reviewers **UMiA** as *R1*, **p1oz** as *R2*, **3Auv** as *R3*, and **7r4h** as *R4* respectively.

[1] Jinsol Lee and Ghassan AlRegib. Gradients as a measure of uncertainty in neural networks. ICIP. 2020

---

### Decision · Program_Chairs · 2021-09-27

**Decision:**

Accept (Poster)

**Comment:**

This paper proposes using the KL divergence between the softmax output and the uniform distribution to detect OOD examples (OOD examples are expected to have lower KL divergence). Authors further show extensive empirical results in support of their method.

All reviewers agree that this work is novel, well-written and is a significant contribution to the community. Reviewers have provided detailed comments to improve this work and increase its impact and I highly recommend authors to take all these suggestions into account. In particular, I suggest further explanation and investigation about the effectiveness of this technique when the class label sets are distinct as well as when there is covariate shift.

Given reviewers' consensus, I recommend accepting the paper.